# Reverse Engineering of Option Pricing: An AI Application

**Bodo Herzog [1,2,]* and Sufyan Osamah [1]**

[1] ESB Business School, Reutlingen University, Alteburgstr. 150, 72762 Reutlingen, Germany; sufyanosamah@gmail.com
[2] RRI Reutlingen Research Institute, 72762 Reutlingen, Germany
* Correspondence: Bodo.Herzog@Reutlingen-University.de; Tel.: +49-(0)7121 271-6031

**Abstract:** This paper studies option pricing based on a reverse engineering (RE) approach. We utilize artificial intelligence in order to numerically compute the prices of options. The data consist of more than 5000 call- and put-options from the German stock market. First, we find that option pricing under reverse engineering obtains a smaller root mean square error to market prices. Second, we show that the reverse engineering model is reliant on training data. In general, the novel idea of reverse engineering is a rewarding direction for future research. It circumvents the limitations of finance theory, among others strong assumptions and numerical approximations under the Black–Scholes model.

**Keywords:** reverse engineering; option pricing; derivatives; genetic algorithm; artificial intelligence; machine learning

**JEL Classification:** G13; G11; C52; C53; C61; C65

---

## 1. Introduction

The most remarkable scientific breakthrough in recent years is the creation of super-human intelligence via deep neural networks in the field of artificial intelligence (AI) (Silver et al. 2016, 2017, 2018; Tian et al. 2019). Additionally, the availability of Big Data is opening a new field of data analytics. Given the enormous scale and complexity of data, a manual human analysis is impossible. Therefore, machine learning (ML) is applied in order to extract the relevant information from data.

We utilize the tools of AI and apply them to finance, particularly on pricing financial derivatives. This paper focuses on a certain financial derivative, namely options. There is a diverse array of options such as American options, European options or Exotic options. With the aim to simplify the subsequent analysis, we first study plain vanilla call- and put-options. A call(put)-option gives the holder the right to buy (sell) an underlying asset at a specified strike price and time in the future. Note, European options can only be exercised at the end of maturity, while American-type options can be exercised at any time.

The theoretical methodology of option pricing follows the well-known Black–Scholes (BS) model (Black and Scholes 1973). Although the model is influential, it is held back by simplified assumptions on stock market dynamics, constant variance, no transaction costs, continuous trading, no-arbitrage and complete markets. Moreover, the Black–Scholes model is sometimes lacking closed-form analytical solutions. Generally, a violation of the assumptions leads to deviations from the market price (Liu and Wang 2013).

In order to circumvent the limitations of Black–Scholes theory, we apply a reverse engineering (RE) approach. In short, we use market data in order to search for an option pricing formula by

utilizing a new AI-methodology. This approach is called 'reverse' because we start with data amid to obtain a pricing formula at the end. We gather the same market input data as for the Black–Scholes model, such as current stock price, strike price, time to maturity, volatility and the risk-free interest rate. Our data contain more than 5615 call-options and 5173 put-options of mainly American-type from the German stock market — in-short DAX. In this paper,we apply so-called genetic algorithms (GA), and optimize the input data with the target to infer a functional relationship that captures the true market price of options.[1]

The application of genetic algorithms is highly effective in search and optimisation problems (Mitchell 1996). Simply put when given data, GA is finding a function that fits the relationship between all inputs and the output with the smallest root mean square error. The inputs are the Black–Scholes input variables and the outputs are the option prices in the market. At first, the GA generates several non-linear functions and calculates via iterations the function with the lowest root mean square error (or mean absolute error). As the GA proceeds action, it adapts towards the global optimal functional-type (Brabazon et al. 2008).

The main findings are: first the option pricing formula of reverse engineering is different to the Black–Scholes solution, particularly because it does not contain a normal probability distribution function. Second, the numerical evaluation of the RE model statistically yields a lower root mean square error to the market price. Nonetheless, the pricing of the RE model is not always more plausible than the Black–Scholes model, especially if we are pricing options dissimilar to our training data. This demonstrates that the RE model is heavily reliant on training data. Indeed, pricing options from other markets lead to insufficient results because the RE-formula is not trained by foreign option characteristics.

No doubt, reverse engineering is a novel methodology in the toolbox of modern finance, in particular for solving complex and dynamic problems. Even though the application of AI is promising, our study reveals that it is not devoid of imperfections.

The paper is structured as follows. Section 2 summarizes the relevant literature. Section 3 describes the data, AI-methodology, and option theory. The results are discussed in Section 4. Section 5 concludes the paper.

## 2. Literature Review

This paper is related to at least two areas in economic literature. First, the finance literature about option pricing. The most widely used formula for calculating the price of an option is the Black–Scholes model (Black and Scholes 1973). The standard Black–Scholes model assumes 'ideal' market conditions. That means no transaction costs, market efficiency, no dividend pay-outs, a constant volatility and a risk-free interest rate as well as stock dynamics that follow a geometric Brownian motion.

According to empirical research, the Black–Scholes model reveals systematic deviation between the theoretical and market price (Aboura 2013). Hence, Merton (1976) extended the model towards empirically realistic jump diffusion models. Similarly, Hull and White (1987) removed the assumption of constant variance by presuming stochastic volatility. Naturally, there are several further extensions of the Black–Scholes model for instance by Heston (1993) towards stochastic volatility. Or, similarly by Cont and Tankov (2004) and references therein, such as Chan (1999) and Carr et al. (2003), towards Lévy processes and pricing with jump processes or related approaches, among others the Variance-Gamma model and Normal-Inverse-Gaussian (NIG) model. Corrado and Su (1996) enhanced the Black–Scholes model by adding abnormal skewness and kurtosis, which is observed in empirical data. These extensions did render the option pricing formula of Black–Scholes more representative, however, the new complexity still entails conceptual and computational limitations.

---

[1]  This type of evolutionary computation is based on an algorithm for global optimisation (finding the minima or maxima of a function), on a given set of data through trial and error. It is inspired by the Darwinian biological evolution.

Second, there is a growing literature on AI and ML. ML is the study of algorithms and statistical methods designed to solve a particular task using inferences or pattern recognition (Bishop 2013). A subfield of AI is the application of genetic algorithms or deep neural networks. Both methods are biologically inspired and they are applied in order to process large and complex data (Van Gerven and Bohte 2017). Given the unpredictable nature of stock market dynamics, the application of AI is advantageous (Metcalf and Malkiel 1994). Liu and Wang (2013) show that the option price can be decomposed in three terms $C = C^{BS} + C^{AR} + C^{DNN} + \epsilon$, where $C^{BS}$ denotes the option price from the Black–Scholes model, $C^{AR}$ the price from an autoregressive model, and $C^{DNN}$ the price from a deep neural network; $\epsilon$ is a standard error term. Liu and Wang (2013) claim that this decomposition leads to more precise pricing than the use of one of the three models.

There is further research capitalizing on AI in option pricing. One of the original attempts with neural networks in option pricing is by Hutchinson (1994) and Chen and Lee (1997). This research sparked more research on using AI, however due to data and methodological limitations this research was waived in the field of finance until recently (Yao et al. 2000). Nonetheless, recent discoveries in AI-techniques have given it a new lease of life in finance research (Andreou et al. 2008; Schmidt and Lipson 2009). Nevertheless, there are decisive implementation boundaries in finance. The AI-methodology is essentially using trial and error and so the processing power and computational time rise exponentially as complexity increases. This means that complicated problems have to be broken down into smaller less complicated ones, thus reducing the applicability of AI (James et al. 2017).

## 3. Methodology, Theory and Data

Genetic programming is a non-deterministic and non-linear methodology that is newly uncovering even symbolic equations from data. The most recent advancement of this technique has been developed to analyse complex laws in physics in 2009. The methodology is mainly inspired by evolution and applicable to any nonlinear dynamical system. We utilize this new type of genetic programming and employ this method on option pricing, particularly fitting non-linear functions between several input data and one output variable (Schmidt and Lipson 2009).[2]

The algorithms work as follows: they split the data into training and validation data. Training data is used to fit the parameters, while validation data is utilized in an unbiased way to assess the model. This separation has several advantages while it prevents overfitting (James et al. 2017). Overfitting can be conducive to a model that is too close to the data, which later limits the generalization of the model (Prechelt 2012). All details on genetic programming and the respective source code is available online.[3] Previous genetic algorithms did not allow to model each variable separately and finally did not extract the best fitting analytical equations automatically.

Intuitively, the genetic algorithm proceeds in the following steps: First, a random population of solutions is generated. Second this set of solutions is creating new generations of solutions. Third in order to produce a new generation, the genetic algorithm executes several sub-procedures. Each solution is scored according to a numerical value; called a 'fitness score' (e.g., the mean absolute error). This is repeatedly done by three steps:

(i)   Selection: this is essentially the unique solution that will be used as a base for the next generation of solutions (offspring);
(ii)  Crossover: this is a rule that dictates how the solution will be combined for the next generation of solutions;
(iii) Mutation: this is a random modification of the solutions.

---

[2]   This is implemented in the software package Eureqa (https://www.nutonian.com/index.php).
[3]   Source code is available on https://github.com/verdverm/go-eureqa. Note: the code consists of several elements and it is uncommon to discuss it in an applied finance paper in detail.

The genetic algorithm proceeds all steps over many sequential generations in order to obtain the global optimum, defined by the smallest mean absolute error or root mean square error. The program of our genetic algorithm can be summarized by the following pseudo-code:

```
START
Generate the initial random population/function (training data)
Compute fitness
REPEAT
 Selection
 Crossover
 Mutation
Compute fitness
UNTIL population/function has converged
STOP
```

We demonstrate the computational power of this algorithm on option pricing. The automated processes for distilling the data into the form of analytical formulas is promising, particularly in the field of complex derivatives. Unlike econometric models, an automated reverse engineering approach with a GA-methodology is free from assumptions and identifies the real-life empirical relationships.

In finance, option pricing is mainly based on Black–Scholes theory. Suppose we have a call option whose value $V(S,t)$ depends on the asset price $S(t)$ and time $t$. The asset price is modelled according to the geometric Brownian motion: $dS(t) = \mu S(t)dt + \sigma S(t)dW(t)$. We assume that markets are complete and there is no-arbitrage. The theory and evidence from behavioral finance demonstrate that those conditions do not hold in general (Kahneman and Tversky 1979; Thaler 1980; Thaler et al. 1997). Although, we know that these assumptions are too strong in reality, they are necessary in order to solve the Black–Scholes model. Thus, there is a pressing scientific application of our reverse engineering methodology in finance.

Our goal is to find option pricing formulas by employing the method of genetic programming. Consequently, our approach differs significantly to the derivation of Black–Scholes. Black and Scholes(-Merton) use Itô's formula on the value function $V(S,t)$ together with further assumptions that $V(S,t)$ is continuously twice differentiable in $S$ and once in time $t$ in order to derive the option price formula. The derivation of the Black–Scholes formula is in Appendix B. We obtain the well-known Black–Scholes equation

$$\frac{\partial V(S,t)}{\partial t} + rS_t\frac{\partial V(S,t)}{\partial S} + \frac{1}{2}\sigma^2 S_t^2\frac{\partial^2 V(S,t)}{\partial S^2} - rV(S,t) = 0. \tag{1}$$

Having derived the 2nd-order partial differential equation above, or in short the Black–Scholes Equation (1), we find the solution for a call- and put-option respectively.

**Theorem 1.** *Black–Scholes Formula for a European call- and put-option is given by*

$$
\begin{aligned}
V(S_t,t) &= S_t N(d_1) - Ke^{-r(T-t)}N(d_2), && \textit{Call option} \\
V_P(S,t) &= P(S,t) = KN(-d_2)e^{-r(T-t)} - S_t N(-d_1), && \textit{Put option} \\
d_{1,2} &= \frac{\ln\left(\frac{S_t}{K}\right) + (r \pm \frac{1}{2}\sigma^2)(T-t)}{\sigma\sqrt{T-t}},
\end{aligned}
\tag{2}
$$

*where $N(x) = \int_{-\infty}^{x}\frac{1}{\sqrt{2\pi}}e^{-\frac{s^2}{2}}ds$ denotes the cumulative standard normal probability distribution.*

There is a nice proof of the Theorem, via a transformation of the Black–Scholes Equation (1) to a heat equation, available by Guenther and Juengel (2003).

The Black–Scholes model is our benchmark for theoretical considerations only. In our paper, we demonstrate how a genetic algorithm identifies a similar analytical function based on a new GA-methodology. We search a call option and a put option function of the form $F(S, K, \sigma, T - t, r)$, where $S$ denotes the stock price, $K$ the strike price, $\sigma$ volatility, $T - t$ time to maturity and $r$ the risk-free interest rate. We choose the model with the highest R-square and lowest root mean square error given computational efficiency. The GA-methodology quantifies this by the term size—another word for complexity. In our approach, we select the model with a size of approximately 40. Our algorithm computes models up to a size greater than 200. The algorithm stops once a certain criterion such as the number of generations or the time the algorithm is running is reached. We set the limit of 100 h until the algorithm cuts off (stopping criteria). The assessment of the best model follows statistical parameters (Willmott and Matsuura 2006). The obstacle of a genetic algorithm is that we are able to find meaningful, nontrivial and general analytical relationships for certain invariants.

Our work is based on option data from 25 different companies from the German stock market index DAX (Table A1). We include both in-money, at-money, and out-of-money options. We download the data from a single day by FactSet (Bakshi et al. 1997). We follow the procedure that is common in the literature, however, deviate in one respect, due to our AI-methodology, and use raw price data. In the end, we create a big data set including all traded options. The maturity date is almost always within 3 months, i.e., 3 month or less. In total, we have data of 5616 call options and 5174 put options. In addition, we gather the present stock prices, strike prices, implied volatilities and the risk-free interest rate. We choose the 3-months German bond as the risk-free interest rate.

## 4. Results and Discussion

The results of the GA-methodology reveal several interesting and new findings. First, the exercise gives us a new call- and put-option function in order to price different options based on our German data. Both functions are automatically derived by evolutionary search from the GA-methodology. Note, the output of the new GA-algorithm are both analytical formulas. Hence, we do not merely estimate single coefficients, indeed, we automatically estimate the whole analytical function by the GA-algorithm in order to obtain a functional output as given next. The functions are

$$C = \frac{S(t) - K(\sigma + (T - t) \times K^2 \times \sigma^2)^r}{\sigma^r - K^{(K - S(t))}} \tag{3}$$

$$P = \frac{K^2}{S(t)} + \frac{K - S(t)}{(T - t)^{1+r} + e^{\left[K + \frac{S(t) \times K}{S(t) - K}\right]}} \tag{4}$$

The computational quality of our option functions is surprisingly high, given the unconventional functional-shape. The call option function has a R-square of 0.956, a correlation coefficient of 0.979, and mean absolute error of 0.533. Similarly, for the put option, we obtain a R-square of 0.969, a correlation coefficient of 0.985 and the mean absolute error of 0.479. Both formulas do not follow a straight forward intuition such as the Black–Scholes equation. However, this is not surprising given that both formulas are obtained from data with an assumption-free genetic algorithm.

Indeed, our formulas are applicable to real market conditions and work in cases of incomplete markets or under the condition of arbitrage. Additionally, the formulas are computed by options with a maturity of 3-month or less, which makes the economic interpretation rather sophisticated. On the one hand, this enhances the predicative power but surprisingly does not limit the pricing of options with different maturities on the other hand. Furthermore, the algorithm does not explain why these formulas are the best fit, except that they both have the lowest mean absolute error. But this is a known disadvantage of modern AI-techniques as discussed in the literature (Silver et al. 2018). Finally, one point is striking: the pricing functions under the RE model do not

contain any normal probability distribution as under the Black–Scholes model (2). This is surprising because the GA-methodology is also searching for error-type functions, which are a functional equivalent to the normal probability distribution.

In that regard, there is the need of further research in order to generalize and explain the output of the GA-methodology. A special focus ought to be on RE models with option formulas with different maturities, the property of no-arbitrage and formulas having a similar form to the Black–Scholes model.

In the following, we study the robustness and sensitivity of our formulas. First, we compare the pricing performance with the Black–Scholes model. Table 1 represents the main results. We check our model with 5615 call options and 5173 put options.

**Table 1.** In-sample assessment of pricing call- and put-option (DAX).

|  | **Option Price** | **MAE** | **RMSE** | **No. Obs** |
|---|---|---|---|---|
| Avg. Market Value call | 8.81€ | - | - | 5615 |
| Avg. BS-Value call | 8.30€ | - | - | 2791 |
| Avg. RE-Value call | 9.00€ | - | - | 5615 |
| Market vs. BS call | 0.51€ ** | 0.685 | 0.368 | 2791 |
| Market vs. RE call | −0.19€ | 0.100 | 0.118 | 5615 |
| BS vs. RE call | −0.70€ *** | 1.511 | 0.871 | 5615 |
| Avg. Market Value put | 4.99€ | - | - | |
| Avg. BS-Value put | 5.31€ | - | - | |
| Avg. RE-Value put | 5.10€ | - | - | |
| Market vs. BS put | −0.32€ *** | 2.323 | 27.825 | 5173 |
| Market vs. RE put | 0.11€ | 0.555 | 4.028 | 5173 |
| BS vs. RE put | −0.70€ *** | 2.245 | 30.102 | 5173 |

Note: ** $=p < 0.05$, *** $=p < 0.01$. Source: authors computations.

The average market price of calls is of 8.81 Euro. The average Black–Scholes call price is of 8.30 Euro, while the reverse engineering mean price is of 9.00 Euro.[4] The mean variance in our RE model is of 0.57 and lower than the Black–Scholes model (0.77). Counting the cases when the RE-option price has a smaller mean absolute error to the market price demonstrates that the RE model is outperforming the Black–Scholes model in 2740 out of 2791 cases or over 98% of time. The mean absolute error (MAE) is of 0.100 between the market price and the RE-option price (Table 1). Similarly, the root mean square error (RMSE) is of 0.118 and it is smallest in the RE model. Furthermore, the null-hypothesis of equality in case of our RE model to the market price cannot be rejected, meaning that the RE model is statistically closely related to the market price of options.

The evaluation of the RE model in regard to put options demonstrate a similar accuracy. For put options the performance of the RE model is robust again. The null-hypothesis of price equality cannot be rejected for the RE model, i.e., our model is significantly better than the Black–Scholes model. Moreover, the MAE of our model is of 0.555 and RMSE is of 4.028. Hence, both statistical measures are the lowest when comparing it with the Black–Scholes model. All in all, the statistical findings corroborate the predictive power of the RE model that is based on the new formulas in Equations (3) and (4).

To test if the RE model works on data dissimilar to the training data, Amazon option data were used for the same maturity as above.[5] In Table 2, we summarize the performance of an out-of-sample assessment of our RE model. We use 147 call-options frequently traded in the US market.

---

[4]　The price is calculated with the BS-Formula for European options or the Binominal-Tree formula for American options.
[5]　We used all available German option data and therefore a robustness check cannot be done with German data. It is neither meaningful nor possible.

**Table 2.** Out-sample assessment of pricing of european call-option (Amazon US).

|  | Option Price | MAE | RMSE | No. Obs |
|---|---|---|---|---|
| Avg. Market Value call | 145.13$ | - | - | 147 |
| Avg. BS-Value call | 149.30$ | - | - | 147 |
| Avg. RE-Value call | 143.49$ | - | - | 147 |
| Market vs. BS call | −4.172$ | 549.371 | 23.439 | 147 |
| Market vs. RE call | 1.637$ | 688.978 | 26.248 | 147 |
| BS vs. RE call | 5.809$ | 96.446 | 9.821 | 147 |

We find that in this scenario, the MAE and RMSE is by far greater than above. Moreover, the computation indicates the RMSE of 26.248 is largest for the RE model (Table 2). The Black–Scholes model has a smaller error despite a mean-price difference of −4.172$ to the market price. Noteworthy, the option price difference across all models is not significant, despite large error terms in comparison to Table 1.

This finding is not surprising. As commonly known in the AI-literature, the quality of the model is heavily reliant on the training data. Our training data consist of German options. Likely this sample does not fit option characteristics in other markets similarly well. This limits the application and generalization of the new formulas to foreign option markets.

Next, we study the sensitivity of our RE model. Firstly, we analyze the sensitivity of the option price in our RE model. The sensitivity in this case is based on calculating the change of one standard deviation (SD). So, a sensitivity of 3 means that if you increase the variable by 1 SD, the output varies by 3 SD. Secondly, we compute the correlation between the respective input variable and the option price. A number of 1.0 means that the input and output variable moves to 100 percent in the same direction.

Two issues stand out in Table 3. First, a change in the risk-free interest rate and maturity have a negligible impact on the option price in our RE model. This is not surprising because in our training data all maturities are within 3 months. Thus, the interest rate is almost always the same—in other words it is a constant. Therefore, the RE model is unable to evaluate a shift in the risk-free interest rate. Second, the other variables denote a significant sensitivity and correlation between input and output. The results are as expected as indicated by our option pricing formula.

**Table 3.** Sensitivity analysis for call- and put-options.

| Variable | Call Sensitivity | Call Correlation | Put Sensitivity | Put Correlation |
|---|---|---|---|---|
| Stock price | 3.68 | 1.00 | 1.56 | 0.33 |
| Strike price | 3.67 | 0.02 | 1.60 | 0.87 |
| Volatility | 0.02 | 1.00 | 0.00 | 0.00 |
| Interest Rate | 0.00 | 0.00 | 0.00 | 0.00 |
| Maturity | 0.02 | 1.00 | 0.06 | 0.91 |

Source: authors computations.

Finally, we study in a scenario analysis the Greek Letters in more detail. Moreover, we employ our formulas to an option with a maturity of 14 days in order to study the predictability and universalizability of our formulas, which are trained by maturities within 3 months. Table 4 represents the results. In the first column, we describe the benchmark case. In the columns to the right-hand side, we study a change of always one variable, while keeping everything else constant (ceteris paribus).

**Table 4.** Sensitivity analysis of greek letters for call-options.

| Variable | Benchmark | $\Delta_S$ | $\Delta_K$ | $\nu$ | $\rho$ | $\Theta$ |
|---|---|---|---|---|---|---|
| Stock price in € ($S$) | 72.16 | $S + 10$ | - | - | - | - |
| Strike price in € ($K$) | 51.00 | - | $K + 10$ | - | - | - |
| Volatility ($\sigma$) | 0.85 | - | - | $\sigma + 0.1$ | - | - |
| Interest Rate ($r$) | $-0.53$ | - | - | - | $r + 0.1$ | - |
| Maturity ($T$) | 14 | - | - | - | - | $T + 1$ |
| RE-price in € | 22.29 | 32.28 | 12.64 | 22.36 | 22.08 | 22.31 |
| BS-price in € | 21.22 | 31.26 | 12.07 | 21.29 | 21.22 | 21.24 |
| Difference RE-BS in € | 1.07 | 1.02 | 0.57 | 1.07 | 0.84 | 1.07 |

Source: authors computations.

We randomly select a call option with the underlying stock price of 72.1 Euro, strike price of 51.0 Euro, volatility of 0.85, and time to maturity of 14 days. Given these parameters the call option price is of 22.3€ if using the RE model and 21.2 Euro if using the Black–Scholes model. Noteworthy, the market price is of 22.1 Euro. Now, we compute the option prices with both formulas by varying the input variables. Table 4 uncovers that the option price of the RE model is always greater than the BS-price and in the range of $[0.57; 1.07]$.

Finally, we analyze the same for the put options. Here, the RE model yields an option price of 31.4 Euro and of 33.1 Euro if using the Black–Scholes model (Table 5). The actual market price is of 30.1 Euro. The sensitive analysis follows the same procedure as for the call option in Table 4. In the case of put options, the RE model yields always a lower price than the Black–Scholes model in a range of $[-1.41; -2.17]$. Of course, all Greek Letters are in line with financial theory.

**Table 5.** Sensitivity analysis of greek letters for put-options.

| Variable | Benchmark | $\Delta_S$ | $\Delta_K$ | $\nu$ | $\rho$ | $\Theta$ |
|---|---|---|---|---|---|---|
| Stock price in € ($S$) | 147.06 | $S + 10$ | - | - | - | - |
| Strike price in € ($K$) | 180.00 | - | $K + 10$ | - | - | - |
| Volatility ($\sigma$) | 0.39 | - | - | $\sigma + 0.1$ | - | - |
| Interest Rate ($r$) | $-0.53$ | - | - | - | $r + 0.1$ | - |
| Maturity ($T$) | 21 | - | - | - | - | $T + 1$ |
| RE-price in € | 31.43 | 22.07 | 40.84 | 31.43 | 31.52 | 31.43 |
| BS-price in € | 33.07 | 23.48 | 43.01 | 33.32 | 33.06 | 33.09 |
| Difference RE-BS in € | $-1.64$ | $-1.41$ | $-2.17$ | $-1.89$ | $-1.54$ | $-1.66$ |

Source: authors computations.

Our reverse engineering approach reveals pros and cons. The major drawback of our approach is its reliance on training data. Secondly, the selection of input variables and the specification of the GA-methodology affects the respective output. In our work, we utilize the same input variables than under the Black–Scholes model. Nonetheless, other input variables may lead to other outcomes. Lastly, the functional output might change if we specify different mathematical operators under the GA-methodology. In our eyes, however, all limitations are excellent topics for further research.

## 5. Conclusions

This paper employs, to our knowledge for the first time, a reverse engineering approach in option pricing. Although the Black–Scholes model is currently the cornerstone in option theory, it is clearly not the end of all of research in option pricing. In theory, there are strong theoretical assumptions that do not always follow the observations in the real world. More recently, the theory in behavioral finance revealed further flaws in the concept of the Black–Scholes theory. Due to the advances in AI and the growing computational power, reverse engineering is expected to gradually grow in all

scientific fields. The methodology employed in this paper allows a new view on those issues in finance. The application of reverse engineering in finance and economics might reveal further surprising facts and theories.[6]

As in physics, the field of finance is embedded with high complexity and dynamics. Thus, reverse engineering reveals new relationships that are not uncovered by a theoretical approach based on simplifying assumptions. The main finding is that the RE model obtains comparable results to the Black–Scholes model. Indeed, the option price of the RE model has even a lower mean absolute error and root means square error than the BS-model. To that extent, the RE model is outperforming the BS-model. However, when using data that is dissimilar to the training data, Black–Scholes outperforms the RE model. Consequently, the RE model creates an inherent bias—sometimes even hidden—towards the training data. But this is a well-known disadvantage of any data mining methodology. Nonetheless, we need further research, particularly using other AI-tools such as deep neural networks. The concept of deep learning with self-learning and self-training mechanisms might even mitigate some biases and the reliance on training data.

**Author Contributions:** For the article the research work was divided as follows: conceptualization and methodology, B.H.; validation, S.O.; formal analysis, B.H.; data curation and data analysis, S.O.; writing, S.O. and B.H.

**Funding:** This research received no external funding. The APC was waived by the editor of the journal.

**Acknowledgments:** We would also like to thank our research assistant Adrián Elsässer Briones for editing and commenting a preliminary version of this paper. We thank the anonymous reviewers for good comments. All remaining errors are our responsibility.

**Conflicts of Interest:** The authors declare no conflict of interest.

**Appendix A. Data**

**Table A1.** Company data from factset.

| Company Ticker | Name | Market Price |
|---|---|---|
| ADS.DE | Adidas | 210.6 |
| BAS.DE | BASF | 66.4 |
| BAYN.DE | Bayer | 69.49 |
| BEI.DE | Beiersdorf | 86.12 |
| BMW.DE | BMW | 72.16 |
| CBK.DE | Commerzbank | 6.63 |
| DAI.DE | Daimler | 50.08 |
| DBK.DE | Deutsche Bank | 7.68 |
| DB1.DE | Deutsche Boerse | 113.7 |
| DEQ.DE | Deutsche EuroShop | 26.58 |
| LHA.DE | Deutsche Lufthansa | 22.13 |
| PBB.DE | Deutsche Pfandbriefbank | 10.58 |
| DPW.DE | Deutsche Post | 27.65 |
| DTE.DE | Deutsche Telekom | 14.86 |
| DWNI.DE | Deutsche Wohnen | 42.56 |
| EOAN.DE | EOAN | 9.73 |
| BOSS.DE | Hugo Boss | 61.56 |
| MRK.DE | Merck | 95.34 |
| B4B.DE | Metro | 14.23 |
| MUV2.DE | Muenchener Rueckversicherung | 208.7 |
| PAH3.DE | Porsche | 56.9 |
| PUM.DE | Puma | 38.2 |
| SIE.DE | Siemens | 95.14 |
| UCG.IT | Unicredit | 11.19 |
| VOW3.DE | Volkswagen | 147.06 |

---

[6] For instance, an analysis of sovereign defaults according to Herzog (2016).

## Appendix B. Derivation of Black–Scholes Formula

First, compute the total derivative of the function $V(S(t); t)$:

$$dV(S,t) = \left[\frac{\partial V(S,t)}{\partial t} + \mu S \frac{\partial V(S,t)}{\partial S} + \frac{1}{2}\sigma^2 S^2 \frac{\partial^2 V(S,t)}{\partial S^2}\right]dt + \sigma S \frac{\partial V(S,t)}{\partial S}dW(t). \tag{A1}$$

Further, they construct a portfolio $\Pi(S,t)$ consisting of buying one option $V(S,t)$ and selling a certain number, $a$, of the underlying asset $S(t)$ such as the portfolio, $\Pi(S,t)$, is risk-free. The portfolio is defined as

$$\Pi(S,t) = V(S,t) - aS(t). \tag{A2}$$

The derivative of the portfolio is given by

$$d\Pi(S,t) = dV(S,t) - adS(t). \tag{A3}$$

Substituting Equation (A1) and the geometric Brownian motion in Equation (A3) yields

$$d\Pi(S,t) = \left[\frac{\partial V(S,t)}{\partial t} + \mu S \frac{\partial V(S,t)}{\partial S} + \frac{1}{2}\sigma^2 S^2 \frac{\partial^2 V(S,t)}{\partial S^2} - \mu a S\right]dt + \sigma S \left[\frac{\partial V(S,t)}{\partial S} - a\right]dW(t). \tag{A4}$$

In order to eliminate the stochastic term, we choose $a$ such as

$$a := \frac{\partial V(S,t)}{\partial S}. \tag{A5}$$

This is called a 'delta–hedge'. The return of the portfolio is now fully deterministic and described by

$$d\Pi(S,t) = \left[\frac{\partial V(S,t)}{\partial t} + \frac{1}{2}\sigma^2 S^2 \frac{\partial^2 V(S,t)}{\partial S^2}\right]dt. \tag{A6}$$

Finally, Black and Scholes use the concept of no-arbitrage. The risk-free portfolio $\Pi$ would be $r\Pi dt$. In other words, the risk-free portfolio follows a deterministic differential equation: $d\Pi(S,t) = r\Pi(S,t)dt$. The right-hand side is substituted to the left-hand side of Equation (A6). Due to the design of a risk-free portfolio and no-arbitrage, we are using the portfolio equation and Equation (A5) in order to obtain the well-known Black–Scholes equation for plain vanilla options

$$\frac{\partial V(S,t)}{\partial t} + rS_t \frac{\partial V(S,t)}{\partial S} + \frac{1}{2}\sigma^2 S_t^2 \frac{\partial^2 V(S,t)}{\partial S^2} - rV(S,t) = 0. \tag{A7}$$

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
