# Peer review of "Reverse Engineering of Option Pricing: An AI Application"

_ijfs, doi:10.3390/ijfs7040068_

Round 1
Reviewer 1 Report
The authors improved the methodology section. They also addressed some of the other issues in the response, but are still not clearly addressed in the paper.
Data source and description
What is the source of the data? Are the prices used in the study mid prices (i.e. (bid+ask)/2)? Is the data filtered as usually done in this kind of studies (e.g. Bakshi et al., 1997)? - clear in the response, absent in paper. Should be added to paper.
Option type
Are you sure that the options you use are European-type and not American-type? – clear in the response that options are American-type, not clear in the paper. Should be clearly stated in paper.
From the response, but not from the paper, I understand that, in fact, BS price is not the price of European options computed with the BS formula, but the price of American options computed with a binomial tree method. This should be also stated in the paper.
Results
"Our methodology does not differentiate between American-type and European-type options." - in the response; this should also be stated in the paper.
But what eqs 9 and 10 are for? European or American options? I guess that since data on American options is used, eqs 9 and 10 are for American options. This should be clarified in the paper.
The formula for puts (eq. 10) does not depend on volatility and this is very strange. The paper should discuss why puts are not influenced by volatility. In this context, how do you compute the sensitivities of put prices with respect to sigma in Table 5 if eq. 10 does not depend on sigma?
If possible, in order to give a better impression on the usefulness of the method, I would suggest to do some out-of-sample analysis also on German data: you could train only on some part of the German data and then use the remaining part for out-of-sample analysis. At the moment, I understand that the entire German data is used for training.
Author Response
Dear Referee,
see our detailed 2nd-response letter attached.
We are looking forward to your feedback!

Reviewer 2 Report
see attached report.

Author Response
Dear Referee,
attached you'll find our detailed response. Thank you very much for your excellent feedback and comments. We are looking forward to your feedback.
General Response to Referee Reports No. 1 and No. 2
First of all, we thank both referees for excellent comments, feedback and suggestions in order to enhance our research paper. We are delighted to declare that we have followed your feedback and herewith resubmit our revised paper.
The response consolidates both referee reports because some feedback was similar and other points were contradictory. Thus, we give a detailed response and explanation to our changes.
Note: Our changes in the revised paper and response letter is always highlighted in red text color.
We are looking forward to your response!
Referee
In the Literature Review (sec. 2), standard references on beyond BS models should be added
(Heston, GARCH, exponential L´evy processes such as Variance Gamma or NIG etc.). The
sentence l.72/73 should be reformulated in my opinion (the statement ”the added complexity
entails increasing computational complexity and ultimately inefficiency” is wrong).
Excellent feedback. Thanks a lot.
We include this literature to our paper according to your suggestion. Moreover, we have clarified the statement!
Referee
Section 3 contains well-known facts about the derivation of the BS equation, that should be
familiar to every finance professional. I would recommend to shorten it and to keep only
the main results.
Response
Thanks a lot. In the first-round referee process, we have obtained the recommendation to explain BS in order to illustrate the limitations of BS-theory in comparison to the benefits of our assumption-free AI-approach. Hence, we cannot delete this part completely because we have to follow the recommendations of all referees to a certain extent. Nonetheless, we agree with you and we’ll streamline this part. In fact, we move the derivation of the BS-formula to an appendix. We only keep in the text the main results of Black-Scholes theory.
Referee
The authors choose call/put options with maturity of 3 months from the German market as
training data, and Amazon options as validation data. It seems not sufficient to conclude: data
from another trading day could give very different results, and I do not understand how the
formulas (9) and (10) can be valid for all T when the training data consist of 3 months options
only. A deeper analysis (other trading days, other data sources like e.g. index options) seems
necessary.
Response
This is an excellent comment. First, let us give a general explanation and response in order to clarify our approach. Secondly, we answer it in more detail.
Our study is focusing on options in the German Stock Market (DAX-30). Based on the availability of data in FactSet, we perform our AI-methodology. The GA-algorithm divides the data always into training data and validation data. Hence, the equations (9) and (10) are global solutions according to both training and validation with our data. New data might give slightly different formulas. But analyzing the sensitive of our formulas for all types of German options with different characteristics (including different time-to-maturity) demonstrate that our formulas are robust and significant as well as demonstrate that they capture the market data better than BS. This result is significant and robust and it holds for all available and traded options; note: our sample includes more than 5000 put and 5000 call options of the German option market – thus a large share of this market. A timeseries study, as you suggest, is a topic of future research and beyond the scope and focus of this paper.
From understanding our AI-approach, it is immediately obvious that the Amazon case-study is not the main validation task in our paper. It is only studied in order to show the limitations of our AI-approach. Moreover, we already check the critical issue you have in mind. Frist, the validation is done within our algorithm based on a large sample; total sample consists of more than 10 000 options. Second, our sample does not only contain options that were maturing in 3 months. Indeed, it includes options maturating within 3months, meaning 3 months or less than 3 months, e.g. 2, 3, 4, 6, 8, 12 weeks etc. Thus, we corroborate the validity of our formulas for options with time-to-maturity less than 3 months as well. Third, we show that our formulas do not price “foreign” option markets (Amazon case-study) as good as the German options – as expected! Thus, we clearly state in our paper, the AI-approach is highly dependent on the (German) data.
Nonetheless, our RE-approach is pricing all (German) options in regard to the market price better than the BS formula. That is and remains a robust and significant finding and the main message of our paper! Reason: Our algorithm trains and validates the formulas (9) and (10) with the RMSE/MAE before it finishes the optimization. That makes the approach powerful in finance (there are similar applications in physics). Overall, you mention a good point, however, our approach does what you expect (testing and validation within a large sample) and the Amazon case-study only is a demonstration of the limitation of our approach and not the validation of our formulas!
Next, we’ll respond to your second point. There are two scientific arguments: a) we do not tell our GA algorithm to discriminate. Indeed, time-to-maturity is varying little as the data is within 3months, meaning 3 months or less. However, we are not applying a simple regression etc. Our algorithm trains the data as multi-dimensional and nonlinear inputs versus the output. Not surprisingly our algorithm gives the time-to-maturity some weight in our formulas (9) and (10) in order to find the global optimum. Note, there are input variables to our algorithm, similar to the Black-Scholes model, and the option price is the output variable. Thereafter, our algorithm searches for the best fitting-function between all inputs and output. There are no assumptions about the relationships of input and output at all. Yet, despite narrow time-to-maturity, our algorithm puts weight on time-to-maturity – as expected.
In fact, coming from finance theory, we/you would expect this dependence because option prices vary with time-to-maturity. Indeed, our GA finds this linkage within our data (despite a “narrow time-to-maturity sample”). But this demonstrates and corroborates the power of our GA-approach. The reason: This algorithm is doing training and validation at the same time and thus splits the data automatically. Additionally, we analyze this issue, you are concerned about, in our paper. We study options with time-to-maturity less than three month and show that our RE-formulas provide significantly better results than BS. Furthermore, we demonstrate (cf Greek Letters) that our RE-formulas replicate most of the common option patterns similar to finance theory; except for ‘nu’. In fact, the issue of volatility in equ. (10) is more concerning and trickier to explain than the time-to-maturity paradox – you have in mind.
In short: this approach uncovers implicitly all/more information from data than you expected (despite a narrow time-to-maturity). This is a great scientific finding too. In the global optimum, time-to-maturity is a part of our pricing formulas – as expected by finance theory. How does the GA-algorithm achieve such a good performance?
So far, we have preliminary thoughts: a) a generic algorithm or neural network obtain a global optimum by fitting weighting factors of highly complex multi-dimensional and nonlinear data (incl. hidden layers). In most AI-studies of today, we cannot explain the how, even if the output is excellent – sometimes the output displays superhuman intelligence (citations in paper; Silver et al.). b) Normal scientific logic of inference follows the relationship from assumption to conclusion. However, we reverse directionality of inference in our paper. Thus, we lable it a reverse engineering approach. We show that it can be scientifically as informative as the standard approach in science of today. No doubt, reverse engineering and AI-methods raise new questions from a classical point of view (cf. literature on explainable-AI). But scientific research proceeds always by utilizing congenial work without ignoring objectivity and differences.
Your recommendations towards the usage of other sources, such as index options is a brilliant idea. However, we are a bit more humbled with our paper and focus on “first evidence” about the potential benefits and limitations of the AI-approach. Moreover, note the number of index options on the DAX is rather small comparing to our large sample. Furthermore, demanding a comprehensive analysis about all issues of reverse engineering and AI, is beyond the scope of a single paper. We do not overestimate our findings and we do not have overconfidence in our results, but the first evidence is scientifically robust and significant – even surprising in some regards.
We are honest and say that all research progresses step-by-step, including our reverse engineering approach in finance. We did a first step in this paper. We have a clear focus in order to demonstrate modern GAs in option pricing in this paper. Step two is up to future research and future papers. We are planning these steps - as you recommend - too. But tackling all in once is unfeasible, given the interesting research questions after step one. In summary, we agree that further research needs to be done. However, we think this should not hinder anyone to publish first-rate significant and robust evidence in a new methodology of reverse engineering in finance.
Referee
If we assume formulas (9) and (10) are true, this means that many prominent features of
the financial theory such as call/put parity or ATM approximations do not hold anymore.
This should be investigated: for instance, how do we delta-hedge portfolios if we assume
formulas (9) and (10) are true?
Response
Excellent feedback and question. Your ideas and suggestions, such as studying the issues of the put-call-parity etc. are valuable and for sure a research project in the future. We highly appreciate your excellent recommendation towards a study of delta-hedge portfolios with eqs. (9) and (10).
At the same time, there is no doubt, that these suggestions are beyond the target and scope of this paper. All suggestions must be studied in more detail, i.e. with a focus on rigor and passion. The scope of our paper is different and the first-round feedback from two referees were very good and proportionate to the scope of our approach. The focus of our study is the introduction of a novel method to the finance literature. Even if we wish, it is almost unworkable to provide a once-for-all comprehensive analysis about reverse engineering and AI in this (first) paper. This demand is unfeasible in any scientific field of today, particularly in AI. We reiterate again: we do not overestimate our findings and we do not have overconfidence, but the first evidence is scientifically robust and significant.

Round 2
Reviewer 1 Report
NA
Author Response
General Response to Referee Reports No. 1 and No. 2
We thank both referees for excellent comments, feedback and suggestions. We are delighted to declare that your comments have enhanced the paper substantially. Herewith re-submit our revised and final paper. Thank you for your help!
Reviewer 2 Report
I appreciate the details provided by the authors in their reply; the issues I raised in my report are globally addressed, and, even if I would have expected a deeper analysis regarding the financial intepretation of the results, I think it is fair to continue the submission process.
I would recommend some corrections in V2 before definitely accepting the manuscript for publication:
l.71 in introduction: Lévy is misspelled (Lvy), and moreover the reference to the Lévy book is inappropriate; I would expect a standard reference on the use of jump models in finance instead (see e.g. Cont and Tankov and references therein);
l.287 Correct "formual" into "formula"
Author Response
General Response to Referee Reports No. 1 and No. 2
We thank both referees for excellent comments, feedback and suggestions. We are delighted to declare that your comments have enhanced the paper substantially. Herewith re-submit our revised and final paper. Thank you for your help!
Response
We highly appreciate the literature reference. We have added it to our literature review accordingly.
Moreover, we have corrected both language issues. First the misspelling of ‘Lévy’, which was caused by missing the apostrophe in LaTeX. The wording is corrected. We also corrected in line 287 the word ‘formula’.
In addition, we did a final proofread of the whole manuscript in order to check the grammar and other possible misspellings after the revision process.
We highly appreciated your comments and help. Thank you!

This manuscript is a resubmission of an earlier submission. The following is a list of the peer review reports and author responses from that submission.
Round 1
Reviewer 1 Report
This paper aims at pricing European call and put options with a reverse engineering approach. The author adopts 25 individual stock options from the German option market to compare the pricing performance of the proposed model with the Black-Scholes model. The author concludes that the proposed model produces smaller pricing errors than the Black-Scholes model.
1. The author should do more empirical analysis on the pricing performance of the proposed model compared to the Black-Scholes model. For example, option prices are very sensitive to the moneyness and maturity. The author should include more stock options with different time-to-maturity and compare the empirical results between the proposed model and the Black-Scoles model across different moneyness groups or different maturity groups.
2. The author compares the pricing performance of the proposed model with the Black-Scholes model and concludes that the former fits the data better. However, the empirical results (Table 1) are not well discussed. For example, the author defines “The row ‘Number of Success’ counts when our computation of the option price is closer to the true market price”. The ‘Number of Success’ should be clearly defined. In addition, it would be better if the author presents the empirical results of absolute pricing error and absolute percentage pricing error which are two well-known criteria to measure the pricing difference between the market and model prices.
3. The author show the a new Call and Put option pricing function based on reverse engineering approach in equations (9) and (10). The author should describe clearly the methodology to produce this new option pricing formula. In addition, the author should provide an economic interpretation of this new option pricing formula.
4. This paper adopts Amazon option data to examine the performance of an out-of-sample assessment of the proposed model (Table 2). The author concludes that if the proposed model works on data dissimilar to the training data, the Black-Scholes model is more accurate. The author should consider giving an explanation of why it is important to employ the market prices of call and put options on the German option market. And it would be better if the author could interpret how to apply the proposed model on different data period of the German option market or on other option markets.
5. Sections 4 & 5: “In total, we have data of 5616 Call-options and 5174 Put-options.” “Table 1 represent the first results. We check our model with 3033 Call-options.” “In Table 2, …We use 149 Call-options frequently traded in the US market .“ The author should define clearly the empirical data such as the sample period and the details of the exclusion filters used to construct the empirical option data.
6. The author only select a call option and a put option as examples to show the sensitivity analysis of the Greek Letters in Tables 4 and 5. Theoretical option prices may vary depends on different level of parameters. The empirical work should be more comprehensive.
Reviewer 2 Report
As far as I know GA is an algorithm to obtain the global maximum, not an algorithm to build functions.
"GA starts with several random functions [..]. If a certain random function has a better fitness then the GA adapts the functional form in this direction" - it is not clear how the adaptation is done
"The option price of the RE-model is even closer to the true option price in the market" - even closer than what?
I think there is no need of section 3. Everybody reading this article is supposed to know the details of BS. Particularly why are we interested in the derivation of the BS valuation equation?
What is really missing is a more detailed description of the methodology employed in the paper.
All the option data are from a single day? It is not clear.
What is the source of the data? Are the prices used in the study mid prices (i.e. (bid+ask)/2)? Is the data filtered as usually done in this kind of studies (e.g. Bakshi et al., 1997)?
Are you sure that the options you use are European-type and not American-type? The authors mention that they use implied volatilities. As we know for European-type options the implied volatilities are obtained by inverting the BS formula: i.e. the implied volatility of a European-type traded option is the value of sigma that one has to input in the BS formula to get exactly the market price of that option. Therefore, one should get a perfect fit for the BS model when using implied volatilities. Not getting a perfect fit suggests that the options used are American-type.
What is small t in eq 9 and 10? Is it, in fact, capital T? There is a strange operation in formula 9 that I am not familiar with: star to the power 2.
Do formulas in eq 9 and 10 satisfy basic no-arbitrage relations? Does the put-call parity hold?
"One point stands out: the pricing functions under the RE-model does not contain any randomness, such as the normal probability distribution under the Black-Scholes model” – I consider this conclusion is not appropriate: the fact that the BS formula contains a function that happens to be the CDF of the normal distribution does not mean that the formula has randomness in it.
As far as I know, single stock options (e.g. options on Amazon) on the US market are of American-type. So this might not be the appropriate data to test a European-type formula on.
References
Bakshi, G., Cao, C., & Chen, Z. (1997). Empirical Performance of Alternative Option Pricing Models. Journal of Finance, 52(5), 2003–2049